# Multifunctional Carbon Dots-Based Fluorescence Detection for Sudan I, Sudan IV and Tetracycline Hydrochloride in Foods

**DOI:** 10.3390/nano12234166

**Published:** 2022-11-24

**Authors:** Min Zhang, Hongmei Yu, Xiaodan Tang, Xiuhui Zhu, Shuping Deng, Wei Chen

**Affiliations:** 1Department of Chemical Engineering, Yingkou Institute of Technology, Yingkou 115014, China; 2School of Chemical Engineering, University of Science and Technology Liaoning, Anshan 114051, China; 3Department of Physics, The University of Texas at Arlington, Arlington, TX 76019-0059, USA

**Keywords:** carbon dots, Sudan I, Sudan IV, tetracycline hydrochloride, fluorescence detection

## Abstract

Sudan dyes are strictly prohibited from being added to edible products as carcinogens and tetracycline hydrochloride (TC) remaining in animal-derived food may cause harm to the human body. Therefore, it is necessary to establish a high-sensitivity, simple and convenient method for the detection of Sudan dyes and TC in foods for safety purposes. In this work, multifunctional blue fluorescent carbon dots (B-CDs) were prepared by a one-step hydrothermal synthesis using glucose as the carbon source. The results show that the fluorescence intensity of B-CDs was significantly affected by the acidity of the solution and can be quenched by Sudan I, IV and TC through selective studies. Interestingly, the fluorescence quenching intensities of B-CDs have a good linear relationship with the concentration of Sudan I and IV at pH = 3–7. The wide range of pH is beneficial to broaden the application of B-CDs in a practical samples analysis. The method has been successfully applied to real food samples of tomato paste, palm oil and honey, and the detection limits are 26.3 nM, 54.2 nM and 31.1 nM for Sudan I, Sudan IV and TC, respectively. This method integrates Sudan dyes and TC into the same multifunctional B-CDs, which shows that the sensor has a great potential in food safety detection.

## 1. Introduction

Sudan dyes are classified as group three carcinogens by the International Agency for Research on Cancer and prohibited from being added to food [1,2]. However, for the sake of visual beauty and making profits, unscrupulous merchants add cheap Sudan I or Sudan IV to ketchup or palm oil to intensify the natural red hue of food and extend the shelf life of the product [3,4,5]. Thus, the detection of Sudan dyes is of great significance in food safety. Tetracycline hydrochloride (TC) is widely used in animal husbandry, especially for the prevention and treatment of bee diseases [6]. Excessive residual or continuous long-term intake of small doses of TC can cause adverse effects such as allergic reactions, gastrointestinal disorders and liver toxicity [6]. Particularly, the residue of TC in honey is an issue that cannot be ignored and is not only a threat to public health but also an obstacle to the international trade of bee products. Therefore, it is of great significance to establish an efficient method for the detection of TC in honey.

At present, there are different techniques to determine Sudan dyes and TC, including chemiluminescence [7], high-performance liquid chromatography [8,9,10], capillary electrophoresis [11,12], fluorescence method [4,6] and so on. Among them, the fluorescence method has attracted wide attention due to its simple operation, low cost, good sensitivity, speed and convenience. Carbon dots (CDs) as a new type of fluorescent probe have good stability, low toxicity, tunable and unique photoluminescence, excellent water solubility and biocompatibility and are widely used in detection [13,14,15,16,17,18] Yang et al. [4] developed sulfur-doped CDs based on the fluorescence resonance energy transfer (FRET) mechanism to detect Sudan I in an acidic solution with a detection limit of 0.12 µM. Zhao et al. [19] prepared an N, P co-doped yellow-emitting CDs based on the internal filtration effect for the detection of Sudan I with a detection limit of 43 nM. Uriarte et al. [20] prepared a kind of CD that could detect TC with a detection limit of 165 nM. Yan et al. [21] prepared CDs for the detection of TC in basic solution based on the internal filtration effect with a detection limit of 6 µM. Although CD-based fluorescence detections of Sudan and TC have been reported in some literature, there are some shortcomings in the preparation process, such as the use of strong acid [4,19] and the need for many raw materials for synthesis [19,20,21]. At the same time, the proposed methods are not sensitive enough, and the detection limits are high [4,19,20,21]. Moreover, in the reported literature, the sensors are all used to detect Sudan dyes or TC, respectively. The integration of the Sudan dyes and TC into one sensor for detection has not been reported in the literature. Therefore, it is significant to develop a multifunctional CDs for detection of Sudan I, Sudan IV and TC.

In this work, a kind of green and cheap blue fluorescent CD (B-CD) was synthesized by a one-step hydrothermal synthesis method only using glucose as the carbon source. A variety of methods was used to characterize the B-CDs, and the optical properties and stabilities of B-CDs were explored. The optical properties of B-CDs under different pH (3–7) and the detection ability of Sudan I, Sudan IV and TC in an acidic environment were studied, and the quenching mechanisms were discussed. Finally, the analytical method was successfully applied to ketchup, palm oil and honey samples for the detection of Sudan I, Sudan IV and TC, respectively. Compared to the reported methods, B-CDs have the advantages of low cost, being green, stability and having a simple preparation process. Simultaneously, this method has a low detection limit and a wide detection range. Furthermore, there are few reports using the fluorescence method to detect Sudan IV. Sudan dyes and TC concentrating on the same CD for detection is reported rarely. Therefore, the B-CDs have great application potentials in food safety monitoring.

## 2. Experimental

### 2.1. Chemicals

The reagents for the metal cation solution, anion solution and amino acid solution were purchased from Sinopharm Chemical Reagent Co., Ltd. (Shenyang, China). Sudan dyes and medicines were purchased from Aladdin Reagent Co., Ltd. (Shanghai, China). Ketchup was Heinz tomato sauce produced by Heinz Foods Co., LTD. (Qingdao, China); Palm oil was super liquid oil produced by Julong Cereals and Oil Co., LTD. (Tianjin, China). Honey was Baihua honey produced by Baihua Apiculture Technology Development Co., LTD. (Beijing, China). All experimental water was ultrapure water. All reagents used in the experiment were analytically pure without further purification. The experiment was carried out in PBS buffer solution (0.1 M).

### 2.2. Synthesis of B-CDs

Glucose (0.5 g) was poured into a polytetrafluoroethylene-lined stainless steel autoclave (25 mL) containing 5 mL ethanol and 5 mL deionized water. Then it was placed into a constant temperature drying oven and heated to 180 °C for 8 h. The brown liquid obtained was filtered with a 0.22 µm microporous membrane to remove impurities and then dialyzed with a dialysis membrane (MW = 500 Da) for 48 h. At the beginning of dialysis, the water in the dialysis bag should be changed frequently and then changed every 4 h until it is clear and transparent. Then the brown B-CD powders were obtained by freeze-drying for 48 h.

### 2.3. Apparatus and Characterization

The morphology and crystalline pattern of B-CDs were performed by a JEM-ARM200F high resolution transmission electron microscope (HRTEM, Hitachi, Tokyo, Japan) and an X’pert pro X-ray diffractometer (XRD, PANalytical, Amsterdam, The Netherlands), respectively. Raman spectrum was obtained by a BX41 spectrometer with a 532 nm laser (Horiba, Kyoto, Japan). The infrared spectra were recorded by an EQUINOX55 FT-IR spectrometer (Bruker, Leipzig, Germany). The XPS spectrum was conducted on an ESCALAB 250 Xi X-ray photoelectron spectroscopy analyzer (Thermo, Waltham, MA, USA). The fluorescence and UV spectra were measured with an FS5 fluorescence spectrophotometer (Edinburgh, Edinburgh, UK) and a U-3900 UV–vis spectrophotometer (Hitachi, Tokyo, Japan), respectively. The fluorescence lifetime was detected by an FluoroMax-4 fluorescence spectrometer (Horiba, Kyoto, Japan). The cyclic voltammetry curve was carried out by a CHI660E electrochemical workstation (Huachen, Shanghai, China).

### 2.4. Experimental Method

First, 0.2 mL B-CDs solution (0.1 mg mL^−1^) and 1.8 mL PBS buffer solution (pH 4.0) were successively added into a quartz cuvette, and then we measured the fluorescence spectra of the B-CDs. Then Sudan I, Sudan IV or TC was added into the B-CD solution, respectively. The corresponding fluorescence spectra were measured at the excitation wavelength of 350 nm after mixing homogeneously and incubating for 2 min, 2 min and 20 s, respectively. The slot widths of the excitation and emission were both 4 nm. Fluorescence titration was used to investigate the sensitivity of B-CDs to Sudan I, Sudan IV and TC, respectively. The interference of coexisting substances that may be contained in ketchup, palm oil or honey was explored. Next, 600 µM of different types of amino acids (L-valine, L-histidine, L-cysteine, L-leucine, L-threonine, glycine, L-phenylalanine and serine), vitamin C, cations (Cu^2+^, Fe^2+^, Fe^3+^, Ca^2+^, Zn^2+^, Mg^2+^, Mn^2+^ and K^+^), anions (H_2_PO_4_^−^, CO_3_^2−^, NO_3_^−^, Cl^−^ and SO_4_^2−^), citric acid, sucrose, and 40 µM dyes (Sudan I, Sudan IV, rhodamine B and neutral red) and medicines (TC, amoxicillin, ibuprofen, roxithromycin, chloramphenicol, oxytetracycline) was added into the B-CD solution, respectively, to evaluate the selectivity and anti-interference ability of this fluorescence sensor. Their fluorescence was measured after mixing for 2 min.

### 2.5. Samples Analysis

The ketchup (340 mg) was dissolved in 3.4 mL ethanol, sonicated for 30 min and filtered with a 0.22 µm microporous membrane to remove insoluble impurities. The above solution was placed in an oven at 100 °C for drying. The obtained solid was dissolved in a PBS buffer solution (pH = 4), and then the sample solution was obtained after filtration using a 0.22 µm microporous membrane [22]. The palm oil (2 g) was transferred into a 50 mL beaker and then heated at 66 ℃ for 10 min in an electric thermostatic drying oven. The heated palm oil (1 mL) was fully shaken and stirred with acetone (9 mL), and the sample solution was obtained after filtering with 0.22 µm microporous membrane [23]. The honey (1 g) was dissolved in 5 mL PBS buffer solution with pH = 4, centrifugated at 12,000 rpm for 5 min and filtered by a 0.22 µm microporous membrane. Then the honey sample solution was obtained [24]. A total of 2 mL B-CD solution (0.1 mg mL^−1^) and the above 500 µL sample solution were transferred to a quartz cuvette, respectively, and then detected according to the experimental method.

In summary, the carbon dots prepared by glucose were used for the determination of Sudan I, Sudan IV and TC in food, as shown in Figure 1.

## 3. Results and Discussion

### 3.1. Characterization of B-CDs

The structure and morphology of B-CDs were analyzed by TEM, XRD, Raman, FT-IR and XPS. As shown in Figure 1a,b, B-CDs nanoparticles have a spherical morphology with an average particle size of 2.76 ± 1.06 nm, which is calculated by counting 160 carbon dots through Image J software. The HRTEM image shows that the lattice fringe spacing of B-CDs is 0.24 nm, which corresponds to the (100) plane of graphitic CDs [25]. XRD was used to further characterize the microstructure of B-CDs. As shown in Figure 1c, a diffraction peak appears at 2θ = 21.15°, corresponding to the diffraction plane (100), indicating that B-CDs have an amorphous carbon structure [26]. Raman spectroscopy was used to further explore the interlayer structure of B-CDs. As shown in Figure 1d, the Raman spectrum of B-CDs shows the D band and G band at 1344 cm^−1^ and 1579 cm^−1^, respectively, and the IG/ID is 1.03. It shows that B-CDs have a high degree of graphitization [25], which is consistent with the TEM and XRD results.

The functional groups on the surface of B-CDs were characterized by FT-IR and XPS. As shown in Figure 2a, 3359 cm^−1^, 2925 cm^−1^, 1702 cm^−1^, 1388 cm^−1^ and 1030 cm^−1^ correspond to the stretching vibration of -O-H [27], -CH_2_ [28], -C=O [27,29], -COO- [30] and -C-O [20], respectively. Figure 2b shows the full XPS spectrum of B-CDs; the peaks at 284.8 eV and 532.42 eV represent C1s and O1s [21], respectively. Among them, the proportion of the C element is 71.48 %, and the proportion of the O element is 28.52 %. Figure 2c is the fitting diagram of the C1s peaks of the B-CDs. The characteristic peaks at 284.58 eV, 285.92 eV and 288.00 eV correspond to C-C/C=C, C-O/C-O-C and C=O groups [21], respectively. Figure 2d is the fitting diagram of the O1s peaks of B-CDs. The characteristic peaks at 532.01 eV and 532.68 eV correspond to C=O and C-OH/C-O-C groups [31], respectively. The results of FR-IT and XPS are basically consistent. It indicates that the surface of B-CDs is rich in -OH, C=O and -COOH groups and confirms that B-CDs have good water solubility [32].

### 3.2. Optical Properties of B-CDs

The optical spectra of B-CDs were measured by using the UV and fluorescence spectrophotometers. As shown in Figure 3a, the UV absorption spectrum of B-CDs shows two peaks at 267 nm and 350 nm, respectively. The absorption peak at 267 nm is due to the π-π* transition [21,33], and the absorption peak at 350 nm corresponds to the n-π* transition of C=O [33]. The excitation wavelength dependence of B-CDs was studied in Figure 3b. It indicates the fluorescence emission spectra of B-CDs are gradually red-shifted with the increase in the excitation wavelength from 300 nm to 420 nm. It displays a typical fluorescence spectrum property of CDs. The fluorescence intensity of B-CDs is the strongest at Ex = 350 nm (pink line in Figure 3b and black line in Figure 3a). At this point, the best emission peak is at 445 nm (blue line in Figure 3a). According to the test method in Reference 16 with quinine sulfate as the reference, the quantum yield of B-CDs measured at 3.0 %.

### 3.3. Fluorescence Properties of B-CDs

(1) Stability

The stability of B-CDs was investigated by studying the effects of pH, ionic strength, oxidant concentration, oxidation time and irradiation time on the fluorescence intensities of B-CDs. As shown in Figure 4a,b, when the pH of the PBS buffer solution was 4.0, the fluorescence intensity of B-CDs was the strongest. Therefore, the following experiments were carried out in pH = 4 PBS buffer solution. Figure 4c shows the fluorescence intensities of B-CDs gradually decreased when the NaCl concentration changed from 0 to 1.0 M. However, the overall decrease was slight, indicating that B-CDs had good salt resistance. When H_2_O_2_ (0–500 µM) was gradually added to the B-CDs solution, the fluorescence intensities of B-CDs remained almost unchanged (Figure 4d). Meanwhile, the fluorescence intensities were detected within 1 h after adding 500 µM H_2_O_2_ to the B-CDs solution. The changes of fluorescence intensity of B-CDs were not obvious within 1 h (inset in Figure 4d), indicating that B-CDs have strong antioxidant properties. The effects of irradiation time under UV (the spectral irradiance of 25 µw cm^−2^) and xenon light (the spectral irradiance of 50 µw cm^−2^) were investigated. As shown in Figure 4e,f, the fluorescence intensities of B-CDs remain unchanged under UV light irradiation within one hour and xenon lamp illumination within two hours. It shows that B-CDs have good photobleaching resistance. Therefore, B-CDs have excellent fluorescence stability and chemical stability.

(2) Selectivity

To explore the application of B-CDs in food safety testing, selective experiments were performed in view of the possible presence of substances in ketchup, palm oil and honey samples. As shown in Figure 5, vitamin C, sucrose, citric acid, five kinds of amino acids (L-threonine, L-leucine, L-histidine, L-cysteine and L-valine ), seven kinds of anions (Cu^2+^, Mn^2+^, Zn^2+^, Fe^2+^, Na^+^, K^+^ and Ca^2+^), four kinds of cations (SO_4_^2−^, H_2_PO^4−^, NO_3_^−^ and CO_3_^2−^), rhodamine b and neutral red have little effect on the fluorescence intensity of B-CDs. However, Sudan I, Sudan IV, TC and Fe^3+^ produced strong quenching responses to the fluorescence intensity of B-CDs. Here, the concentrations of Sudan dyes and TC are 40 µM, and the concentrations of other compounds or ions are 600 µM. It indicated that B-CDs have good selectivity and other substances or ions except for Fe^3+^ do not interfere with the detection of Sudan I, Sudan IV and TC. The interference of Fe^3+^ ions can be eliminated by reducing them to Fe^2+^ ions.

(3) Incubation time

The incubation time of Sudan I, Sudan IV and TC for their reactions with B-CDs was explored. As shown in Figure 6a–c, 20 µM Sudan I, Sudan IV or TC was added to the B-CDs solution, and the fluorescence intensity of B-CDs was rapidly quenched within 1 min for Sudan I and Sudan IV and 20 s for TC, respectively. Subsequently, when the incubation time was 2 min for Sudan I and Sudan IV and 20 s for TC, the fluorescence intensity of the mixed solution tended to be stable and remained basically unchanged within 10 min. It indicated that B-CDs could rapidly, efficiently and stably detect Sudan I, Sudan IV and TC.

### 3.4. Detection of Sudan Ⅰ Based on B-CDs

(1) Linear range and detection limit

B-CDs in the PBS buffer solution (pH = 4) were applied to the detection of Sudan I as shown in Figure 7a. The fluorescence intensity of B-CDs is gradually quenched with the increasing concentration of Sudan I. There is a good linear relationship between F_0_/F and the concentration (0.088–65 µM) of Sudan I (F_0_ and F represent the fluorescence intensity of B-CDs without and after the addition of Sudan I, respectively) shown in Figure 7b. The detection limit of this method was 26.3 nM according to the 3σ rule [21] (LOD = 3σ/k, where σ is the relative standard deviation of the fluorescence intensity of 11 repeated measurements of the blank solution, and k is the slope of the standard curve of F_0_/F versus Sudan I concentration).

To prove the universality of B-CDs for detection of Sudan I in the range of pH = 3–7, the same experiments were carried out in PBS buffer solutions of pH = 3, 5, 6 and 7, respectively. As shown in Figure 8a–h, all have a good linear relationship. Moreover, the emission peaks of B-CDs were unchanged under different pH values. The detection ranges of B-CDs for Sudan I at pH = 3, 5, 6 and 7 are 0.023–55 µM, 0.024–55 µM, 0.022–55 µM and 0.025–45 µM, respectively. It indicates that B-CDs have good universality for the detection of Sudan I at pH = 3–7 and can detect Sudan I in actual samples with different pH value. Compared with the reported fluorescence analysis methods for the detection of Sudan I (Table 1), the sensor has better sensitivity and a wider detection range. In addition, new emission peaks appear at wavelengths longer than 550 nm after the interaction of Sudan I with BCD, which is a new and interesting phenomenon. More investigation is in progress in order to figure out the attributions of these new emissions.

(2) Pigment interference

Ketchup contains lycopene, which may interfere with the determination of Sudan I. A comparative experiment was conducted. Lycopene (0–65 µM) was gradually added to the B-CD solution as shown in Figure 9a. Lycopene can also quench the fluorescence of B-CDs, although its effect is relatively small compared with that of Sudan I. Figure 9b clearly shows the results of this control experiment. The quenching of the fluorescence intensity of B-CDs by lycopene was only 9.9%, which is much lower than the quench by Sudan I (55.6%). The Stern–Volmer equation F_0_/F = 1 + *K_SV_*[*Q*] [16,35] was used to calculate *K_SV_* of B-CDs after adding lycopene or Sudan I, respectively, where F_0_ and F are the fluorescence intensities without and after adding lycopene or Sudan I, respectively. [*Q*] is the concentration of lycopene or Sudan I, and *K_SV_* is the quenching constant. The Stern–Volmer plots of F_0_/F versus the various concentrations of lycopene or Sudan I illustrate a good linear correlation. The quenching constants (*K_SV_*) are 19,239.0 L mol^−1^ and 1694.3 L mol^−1^ for Sudan I and lycopene, respectively, which was 11.4 times that of the latter. Therefore, the interference of lycopene on the detection of Sudan I by B-CDs can be deducted by background deduction.

### 3.5. Detection of Sudan Ⅳ Based on B-CDs

Sudan IV can effectively quench the fluorescence of B-CDs (pH = 4), as shown in Figure 10a. There is a good linear relationship between F_0_/F and the concentration of Sudan IV in the range of 0.18–150 µM, as shown in Figure 10b. The detection limit is 54.2 nM, calculated according to the 3σ rule. Interestingly, Sudan IV can also effectively quench the fluorescence of B-CDs at pH = 3, 5, 6 and 7 (Figure 11). The linear ranges are 0.18–120 µM, 0.29–120 µM, 0.10–120 µM and 0.11–120 µM, respectively, at the same emission peak position. It shows that the sensor has good pH universality, and the method can detect Sudan IV in real samples under various pH environments.

### 3.6. Detection of TC Based on B-CDs

The application of B-CDs in food safety detection was further explored. As shown in Figure 12, B-CDs were found to be particularly sensitive to TC among the six drugs (amoxicillin, ibuprofen, roxithromycin, chloramphenicol, oxytetracycline). When TC was added into the B-CDs solution (pH = 4), the fluorescence intensity of B-CDs was quenched (Figure 13a,b), and the linear range is 0.1–30 µM (Figure 13b (inset)). The detection limit of B-CDs detection for TC is 31.1 nM according to the 3σ rule. The comparison of TC analysis performance with other fluorescence methods reported in the literature is listed in Table 2. This sensor has better sensitivity for TC with a wider detection range.

### 3.7. Mechanism Discussion

Fluorescence quenching mechanisms mainly include fluorescence resonance energy transfer (FRET), photoinduced electron transfer (PET), inner filter effect (IFE), dynamic quenching and static quenching. Among them, FRET and IFE require a certain degree of overlap between the excitation or emission spectrum of B-CDs and the absorption spectrum of the quencher (that is, the excitation or emission light of the fluorescent substance is absorbed by the quencher, so that the fluorescence intensity of the fluorescent substance is weakened). The fluorescence-quenching mechanism of B-CDs in the detection of Sudan I, Sudan IV and TC was determined by fluorescence spectrum, ultraviolet absorption spectrum, fluorescence decay curve and cyclic voltammetry. As shown in Figure 14a–c, the UV absorption spectra of Sudan I, Sudan IV and TC partially overlap with the excitation and emission spectra of B-CDs, indicating that the fluorescence quenching mechanism of B-CDs may be caused by FRET and IFE. However, as seen from Figure 7a, Figure 10a and Figure 13a, no red shift was observed in their fluorescence as the quench agent concentration was increased. Therefore, the quenching mechanism of B-CDs can exclude IFE. Thus, we may conclude that the quenching mechanism of B-CDs is solely due to FRET, which is consistent with the mechanism of Sudan I reported in [4].

The spectrum of the fluorescence decay curve is shown in Figure 15. The average fluorescence lifetime is calculated according to the empirical Formula (1) [40]:(1)$$\tau =\frac{A_{1}t_{1}^{2}+A_{2}t_{2}^{2}+A_{3}t_{3}^{2}}{A_{1}t_{1}+A_{2}t_{2}+A_{3}t_{3}}$$

The average fluorescence lifetime of B-CDs was 3.75 ns, and after the addition of 20 µM Sudan I, Sudan IV or TC, the average fluorescence lifetime became 3.81 ns, 3.83 ns and 3.81 ns, respectively. It can be seen that the average fluorescence lifetime of B-CDs is slightly changed after adding quencher. Meanwhile, it can be seen from Figure 7b, Figure 10b and Figure 13b that the quenching curves are all curved “upward” with the increasing concentration of Sudan I, Sudan IV and TC. This is a clear indication of the presence of static and dynamic quenching.

In order to verify whether there is electron transfer between B-CDs and Sudan Ⅰ, Sudan Ⅳ or TC, the highest occupied molecular orbital (HUMO) and lowest unoccupied molecular orbital energy levels of B-CDs were calculated using the following empirical Formulas (2) and (3) [21]:*E_HUMO_* = −*e* (*E_OX_* + 4.4)(2)
*E_LUMO_* = −*e* (*E_red_* + 4.4)(3)

The cyclic voltammetry curves of B-CDs were measured in the solution state. *E_OX_* represents the oxidation potential of B-CDs. As shown in Figure 16a, the reduction potential *E_red_* of B-CDs is 0.043 V, and *E_LUMO_* was calculated to be −4.44 eV. *E_HUMO_* was calculated by Equation (4) [21]:*E_HUMO_* = *E_LUMO_* − *E_g_*(4)

As shown in Figure 16b, the optical band gap *E_g_* was calculated to be 3.92 eV from the absorption edge of the absorption spectrum of B-CDs, so the *E_HUMO_* value of B-CDs was calculated to be −8.36 eV.

According to [21,32], the *E_HUMO_* and *E_LUMO_* of Sudan I and TC are −5.62 eV and −3.66 eV, −6.2 eV and −3.87 eV, respectively. As shown in Figure 17, the LUMO energy levels of Sudan Ⅰ and TC are higher than those of B-CDs, so the photoexcited electrons of B-CDs cannot be transferred to Sudan I or TC. The Sudan IV solution was found to have no electrochemical reaction. Therefore, no PET could occur between B-CDs and Sudan I, Sudan IV or TC. However, an electron transfer from Sudan I and TC HOMO, respectively, to the B-CD HOMO after photoexcitation of B-CDs can take place to form a B-CD radical, which will be quenched. In Figure 13b, the quenching curve is bent “upward”, indicating the increasing concentration of TC. This is a clear sign that both static and dynamic quenching in the presence of TC occurs. In summary, fluorescence quenching mechanisms of B-CDs by Sudan I, Sudan IV and TC are complex and are the result of synergistic action of various mechanisms.

### 3.8. Application of B-CDs in Actual Samples

The standard addition and recovery experiments were used to test the actual samples to verify the reproducibility and accuracy of the method. Ketchup, palm oil and honey were selected as actual samples for detection of Sudan I, Sudan IV and TC, respectively. As shown in Table 3, the above substances were not detected in the actual samples. Sudan I standard solutions (3 µM, 5 µM and 15 µM) were added to the treated ketchup, and the recoveries are 92.61–96.02%, with the RSD between 0.31% and 0.63%. Sudan IV standard solutions (10 µM, 20 µM and 30 µM) were added to the treated palm oil, and the recoveries are 97.45–101.75%, with the RSD between 1.78% and 2.74%. TC standard solutions (4 µM, 10 µM and 15 µM) were added to the treated honey, and the recoveries are 90.31–109.48%, with the RSD between 0.42% and 0.93%. The above results show that the prepared method has the advantages of accuracy, reliability and strong reproducibility and has the ability to detect Sudan I, Sudan IV and TC in actual samples.

## 4. Conclusions

Blue fluorescent carbon dots (B-CDs), which are environmentally friendly, inexpensive and simple to prepare and have good stability, were synthesized by one-step hydrothermal synthesis with glucose as the carbon source. B-CD spherical nanoparticles have an amorphous carbon structure; the surface is rich in -OH, C=O and -COOH groups; and they have good water solubility. Based on FRET, static and dynamic quenching mechanisms, the B-CDs exhibit good fluorescence quenching responses to Sudan I, Sudan IV and TC. The B-CDs have a wide linear range, low detection limit and good precision and accuracy to detect Sudan I, Sudan IV and TC in food samples.

## Data Availability

Not applicable.

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
