# Peer review of "Multifunctional Carbon Dots-Based Fluorescence Detection for Sudan I, Sudan IV and Tetracycline Hydrochloride in Foods"

_nanomaterials, 2022, doi:10.3390/nano12234166_

Round 1
Reviewer 1 Report
See attached pdf file.

Author Response
Reviewer-1
- It might be good to show the chemical structures of Sudan I, IV, and TC.
Response: Thanks! we have added the chemical structures of Sudan I, IV, and TC in Schematic 1.
- p.5, line 180: it must read “… the when the pH of the PBS buffer …”
Response: Thanks! we have corrected this error.
- p.5, lines 189-192: Please provide values for the spectral irradiance used for the bleaching experiments with the UV and Xenon source.
Response: We have given the spectral irradiance of the UV and Xenon lamps in the new manuscript.
- Fig. 5: I guess the data in Fig. 5 are determined from measurements as in Fig. 6? Please clarify.
Response: Yes. A detailed description is given in the experimental method.
- Fig. 7: Here the authors determine the detection limit to be ~26 nM based on a 3σ criterion. The measurements, however, were done in the range of 88 nM – 65 μM. To verify the limit, it might be good to actually show fluorescence at around 26 nM.
Response: Thanks. The detection limit and the lower limit of the linear range are two different concepts. The detection limit is the lowest concentration that can be measured in theory, and it's defined as 3σ/k. The lower limit is the lowest concentration that can be measured in practice, and it's defined as 10σ/k.
- p.7, line 242: The detection ranges do certainly not start at 0. Please provide the lowest concentrations used in the measurements (similar on p.9, line 272 and 275).
Response: We have given the lowest concentrations of all detection ranges.
- 8, line 266: The authors state that “the interference of lycopene on the detection of Sudan I by B-CDs is negligible”. This statement is certainly not true: Assume that you have a sample with around 80 μM lycopene and you want to detect trace amounts of Sudan I, close to the detection limit (26 nM). Then the quenching will be certainly entirely dominated by lycopene. I suggest to phrase this sentence above a bit more careful.
Response: Thanks. This statement has been modified.
- 11/12: I am not convinced by the discussion of the quenching mechanism for several reasons: excluding dynamic quenching by just one single lifetime measurement (one concentration, Fig. 15) for each quencher is not valid, a concentration series as for the fluorescence spectra must be done.
Response: Thanks. The statement about the quenching mechanism has been modified according to your suggestion.
- How do you make sure that the fluorescence in Fig. 15 is from B-QDs, and is not superimposed by that of Sudan I and IV and TC? There is strong overlap between the absorptions (see Fig. 14), so you will always excite the B-QDs and the quenchers.
Response: The fluorescence spectra of Sudan I, Sudan IV and TC are shown in Figures below. As can be seen from these measurements, the emission peaks of Sudan I, Sudan IV and TC are at 588 nm, 398 nm and 600 nm for Ex=350 nm excitation, respectively. Although the sudan dyes and TC are fluorescent, their fluorescence spectra are not overlapped with that of B-CDs (Em=445 nm) except for Sudan IV. As can be seen from the figure 10a, the addition of Sudan IV did not change the shape of the fluorescence spectrum and the position of the emission peak of B-CDs, but only affected the quenching of BCDs.
- In Fig. 13b the quenching curve is bent “upwards” for increasing concentration. This is a clear sign that both static and dynamic quenching in the presence of TC occurs.
Response: Thanks. This statement has been modified.
- Based on the energy levels in Fig. 17, after photoexcitation of B-CD an electron transfer from Sudan I and TC HOMO, respectively, to the B-CD HOMO can take place to form a B-CD radical, which will be quenched.
Response: Thanks. The statement about the quenching mechanism has been modified according to your suggestion.
- At least at very low (nM) concentration I would immediately exclude the inner filter effect as mechanism, this simply too weak. Also at higher concentrations I do not see any indication if this effect, since there is no systematic spectral red-shift in the fluorescence spectra with increasing quencher concentration.
Response: Thanks. The statement about the quenching mechanism has been modified according to your suggestion.
Reviewer 2 Report
Work requires some adjustments. My suggestions below:
How was the statistical distribution shown in Figure 1a constructed? Fig.1c is not an XRD image, maybe a diffraction pattern
Does the ID / IG ratio represent the ratio of integral values? If so, then how were these spectra deconvolved?
In literature you can find a number of empirical approaches on the degree of graphitization based on the intensity ratio of the defect-induced Raman D and bulk G band [10.1063/1.1674108, [10.1111/j.1151-2916.1978.tb09219.x],c[10.1038/NNANO.2013.4620]] 10.1103/PhysRevB.61.14095], 0.1016/j.carbon.2019.12.094
Fig. 2c and d. How was the background removed? What computer software was used to deconvolve the XPS spectra
Are you sure you need Rys3b in connection with Fig.3a and the comment in the text?
The Y axis in Fig.4 should be described as Counts? Please change the description. The same is true for Fig. 6
Fig.7b- how was the F0 / F ratio determined? Also applies to Fig. 8 and 9, 10, 11
Author Response
Reviewer-2
How was the statistical distribution shown in Figure 1a constructed? Fig.1c is not an XRD image, may be a diffraction pattern
Response: Thanks. Fig 1a is a particle size distribution map which is obtained by the statistical analysis of 160 nanoparticles randomly by Image J software. Fig.1c was measured using the X’pert pro X-ray diffractometer (XRD, PANalytical, Netherlands).
Does the ID / IG ratio represent the ratio of integral values? If so, then how were these spectra deconvolved?
In literature you can find a number of empirical approaches on the degree of graphitization based on the intensity ratio of the defect-induced Raman D and bulk G band [10.1063/1.1674108, [10.1111/j.1151-2916.1978.tb09219.x],c[10.1038/NNANO.2013.4620]] 10.1103/PhysRevB.61.14095], 0.1016/j.carbon.2019.12.094
Response: Thanks. We have carefully read these literatures. In Fig.1d, 1344 and 1579 represent the peak locations of the D and G bands, not the intensity. The relative intensity ratio of D-band to G-band (ID/IG) can be used to characterize the degree of disorder or graphitization of carbon materials. IG/ID was calculated to be 1.03. It shows that B-CDs have a high degree of graphitization. In our new manuscript, Fig.1d has been replaced.
Fig. 2c and d. How was the background removed? What computer software was used to deconvolve the XPS spectra
Response: The background was removed by “Subtract Straight Line”. XPS peak was used to deconvolve the XPS spectra software.
Are you sure you need Rys3b in connection with Fig.3a and the comment in the text?
Response: In the new manuscript, Fig 3a and 3b were redescribed in Page 5, Line 181-186.
The Y axis in Fig.4 should be described as Counts? Please change the description. The same is true for Fig. 6
Response: In the fluorescence spectrum, the fluorescence intensity can be expressed as single photon count (unit: Counts), which is provided by the fluorescence instrument used in our experiment. Therefore, the Y-axis can be described as the fluorescence intensity in our article.
Fig.7b- how was the F0 / F ratio determined? Also applies to Fig. 8 and 9, 10, 11
Response: In Fig 7b, 8, 9, 10, and 11, F0 and F are the fluorescence intensities of B-CDs without and after adding Sudan I, lycopene, Sudan IV or TC.
Reviewer 3 Report
The authors report the preparation of blue-emitting carbon dots (B-CDs) using glucose as precursor and their use as fluorescent probes for the detection of Sudan I, Sudan IV and tetracycline. The manuscript is not of high novelty but the work was well conducted and the results are of interest. The following comments should be considered by the authors :
- experimental. 1) paragraph 2.3 : indicate the laser used for recording the Raman spectra. 2) paragraph 2.4, line 108-110: please clarify : fluorescence spectra of Sudan dyes and TC or fluorescence of B-CDs in the presence of Sudan dyes and TC. Moreover, Sudan dyes and TC are fluorescent. Does their fluorescence interfere with that of B-CDs ?
- I am surprised by the very small diameter determined for B-CDs (0.8 +- 0.2 nm). The authors must carefully check these results which are not in agreement with the TEM image (figure 1b).
- figure 3 and the related text : provide the PL QY of B-CDs.
- figure 5 : the PL quenching observed with Sudan I, Sudan IV and TC is relatively weak. Did the authors try to increase the concentration of these compounds ?
- figure 7 : please comment on the shoulder (or the second signal) observed at ca. 550 nm. Same comment for figure 8.
- the manucsript contains a few typing errors and must be carefully checked by the authors.
Author Response
Reviewer-3
The authors report the preparation of blue-emitting carbon dots (B-CDs) using glucose as precursor and their use as fluorescent probes for the detection of Sudan I, Sudan IV and tetracycline. The manuscript is not of high novelty but the work was well conducted and the results are of interest. The following comments should be considered by the authors :
- experimental. 1) paragraph 2.3 : indicate the laser used for recording the Raman spectra. 2) paragraph 2.4, line 108-110: please clarify : fluorescence spectra of Sudan dyes and TC or fluorescence of B-CDs in the presence of Sudan dyes and TC. Moreover, Sudan dyes and TC are fluorescent. Does their fluorescence interfere with that of B-CDs ?
Response: 1) Raman spectra (from 500 to 2000 cm−1 ) were excited by 532 nm laser, which was added in our new manuscript in Page 3,Line 100. 2) Paragraph 2.4 has been modified. The fluorescence spectra of Sudan I, Sudan IV and TC are shown in Figure below. As can be seen from these pictures, the emission peaks of Sudan I, Sudan IV and TC are at 588 nm, 398 nm and 600 nm, respectively. Although the sudan dyes and TC are fluorescent, their fluorescence spectra except for Sudan IV are not overlapped with that of B-CDs (Em=445 nm). As can be seen from the figure 10a, the addition of Sudan IV did not change the shape of the fluorescence spectrum and the position of the emission peak of B-CDs, but only affected the quenching of BCDs.
- I am surprised by the very small diameter determined for B-CDs (0.8 +- 0.2 nm). The authors must carefully check these results which are not in agreement with the TEM image (figure 1b).
Response: Thanks! we have made modifications in the manuscript according to your suggestion. The average particle size and particle size distribution of 160 carbon dots particles were recalculated. The average particle size of B-CDs is 2.76±1.06 nm.
- figure 3 and the related text : provide the PL QY of B-CDs.
Response: Thanks! Our fluorescence instrument is not capable of measuring the quantum yield.
- figure 5 : the PL quenching observed with Sudan I, Sudan IV and TC is relatively weak. Did the authors try to increase the concentration of these compounds ?
Response: In Figure 5, the concentrations of Sudan I, Sudan IV and TC are all 40 μM, and the concentrations of other interfering substances are 15 times of sudan dyes and TC, namely, 600 μM. So you see the PL quenching observed with Sudan I, Sudan IV and TC is relatively weak. But It can be seen from the Figure5, the selectivity and anti-interference of B-CDs are better. The sensitivity experiments (Figure 7,10, and13) show that the quenching effect is better with the concentration increasing of Sudan I, Sudan IV and TC. When the concentrations of Sudan I, Sudan IV and TC increase to 65 μM, 150 μM and 80 μM, respectively, the fluorescence quenching of B-CDs will be half or less. This indicates that B-CDs is quenched significantly with increasing concentration of Sudan I, Sudan IV and TC.
- figure 7 : please comment on the shoulder (or the second signal) observed at ca. 550 nm. Same comment for figure 8.
Response: With the increasing concentration of Sudan I, B-CDs and Sudan I might form a new fluorescence substance which emitted at 550 nm.
- the manuscript contains a few typing errors and must be carefully checked by the authors.
Response: Thanks! We have checked the entire manuscript carefully and corrected the typing errors.

Round 2
Reviewer 1 Report
I appreciate the authors' efforts and I am generally happy with the revisions made. Yet, two minor corrections should be made:
1. p.6: The spectral irradiances given are not spectral irradiances (which have units of W/(m^2 nm). For the UV source the irradiance is given, while for th eXenon lamp the "irradiance" in counts is not helpful. Please provide the irradiance or the flux (power).
2. p.12: "the fluorescence quenching mechanism of B-CDs can exclude IFE, which is presumed to be FRET[4]: First, the inner filter effect has nothing to do with FRET. FRET would be associated with a decrease of lifetime (which the authors excluded), whereas the IFE would lead to a spectral red-shift with increasing concentration, whcih I do not see in the data.
Author Response
I appreciate the authors' efforts and I am generally happy with the revisions made. Yet, two minor corrections should be made:
- 6: The spectral irradiances given are not spectral irradiances (which have units of W/(m^2 nm). For the UV source the irradiance is given, while for th eXenon lamp the "irradiance" in counts is not helpful. Please provide the irradiance or the flux (power).
Response: Thanks! We have provided the spectral irradiance of the Xenon lamp in the new manuscript: 50 μw cm-2
- p.12: "the fluorescence quenching mechanism of B-CDs can exclude IFE, which is presumed to be FRET[4]: First, the inner filter effect has nothing to do with FRET. FRET would be associated with a decrease of lifetime (which the authors excluded), whereas the IFE would lead to a spectral red-shift with increasing concentration, whcih I do not see in the data.
Response: Thanks! we have made modifications in the manuscript according to your suggestion.
Reviewer 3 Report
Most of the corrections were done by the authors.
- I don't understand why the authors are unable to measure the PL QY of CDs. They should use as reference dye like fluoresceine and compare the PL intensity after excitation at the same wavelength.
- the authors didn't clearly answer about the second signal appearing at ca. 550 nm. If a new compound appears as suggested, a structure should be proposed and confirmed.
- the language can still be improved, especially in the text added by the authors.
Author Response
- I don't understand why the authors are unable to measure the PL QY of CDs. They should use as reference dye like fluoresceine and compare the PL intensity after excitation at the same wavelength.
Response: Thanks! We measured the PL QY of B-CDs which is 3.0% and is given in the new manuscript.
- the authors didn't clearly answer about the second signal appearing at ca. 550 nm. If a new compound appears as suggested, a structure should be proposed and confirmed.
Response: This is a new phenomenon. At this moment, we could not figure out what is the attribution for this new emission yet. We will investigate more about this phenomenon.
- the language can still be improved, especially in the text added by the authors.
Response: Thanks! We have checked and corrected the English in the entire manuscript carefully.
Please check that all references are relevant to the contents of the manuscript.
Response: All references are relevant to the contents of the manuscript.